# A Framework for Integrating Agriculture in Urban Sustainability in Australia

**Arif H. Sarker [1,*,†], Janet F. Bornman [2,†] and Dora Marinova [1,†]** 

[1]   School of Design and the Built Environment, Curtin University Sustainability Policy Institute, Kent Street, Bentley, Perth 6102, Australia; D.Marinova@curtin.edu.au

[2]   Food Futures Institute, Murdoch University, Murdoch 6150, Australia; janet.bornman@murdoch.edu.au

\*   Correspondence: arif.sarker@student.curtin.edu.au; Tel.: +61-432-145-724

†   These authors contributed equally to this work.

**Abstract:** Rapid urbanisation all over the world poses a serious question about urban sustainability in relation to food. Urban agriculture can contribute to feeding city dwellers as well as improving metropolitan environments by providing more green space. Australia is recognised as one of the most urbanised countries in the world, and achieving urban sustainability should be high on the policy and planning agenda. A strong consensus exists among policymakers and academics that urban agriculture could be a tenable way of enhancing urban sustainability, and therefore, it should be a vital part of planning processes and urban design as administered by local and state governments. However, in recent decades, planning has overlooked and failed to realise this opportunity. The most significant constraints to urban agriculture are its regulatory and legal frameworks, including access to suitable land. Without direct public policy support and institutional recognition, it would be difficult to make urban agriculture an integral part of the development and planning goals of Australian cities. Developing and implementing clear planning policies, laws and programs that support urban agriculture can assist in decreasing competing land demands. This study analyses the policy and planning practices that can support integrating urban agriculture into city land-use planning. It examines current practices and identifies existing opportunities and constraints. An integration framework for urban agriculture for Australian cities is presented. If implemented, such a conceptual framework would allow improved sustainability of cities by bringing together the advantages of growing food within a greener urban environment.

**Keywords:** urbanisation; land-use planning; urban agriculture; integration; framework; sustainability

## 1. Introduction

Following the recent unprecedented urban growth, it is now expected that 70% of the world population will live in cities by 2050 [1]. Rapid urbanisation has brought many challenges to cities all over the world, with an uncertain future due to population growth, a rapidly changing climate and increased frequency of extreme weather events, water shortages, energy limitations and food security issues [2]. Urban food security is presenting additional challenges due to ever-increasing rural to urban migration, environmental impacts of broad-acre commercial farming and general access to food that is nutritious and safe. Food security and nutrition are important within any city context, including for high-, middle- and low-income countries [3]. According to Morgan [4], food security strongly impacts economic development but also other national priorities, such as public health, social equity, land use and domestic security. Many have argued that within the urban environment, food security depends largely on access, which is highly influenced by flows of people and energy, city infrastructure,

demographics and policies [5–7]. The issue about feeding urban populations, however, is much more complex, as it is also related to how we use the land in the city.

Policy addressing food security challenges through protecting agricultural land is now a focal point on the international agenda, as many cities in the world are forming tighter legislation to safeguard the highly fertile areas in their rural hinterlands. To enhance urban resilience when faced with change and uncertainty, they are now facilitating a range of urban food production opportunities within their perimeters, such as community-supported agriculture, verge gardening, city farms, food cooperatives, farmers' markets, etc. [8]. Urban agricultural policies incorporated into the planning processes can help to minimise the threats to food supplies from climate change, availability of land, water and oil, as well as having the potential to strengthen regional economies and provide more green space [9]. As such, food production should rightly be seen as an urban development concern. Well-functioning urban food systems are an important part of enabling citizen well-being and functional cities [10].

However, in recent decades, urban planning no longer focusses on food, mainly as a result from the shift from local to global production [11]. The urban (city or local) food system appears to be absent from the planners' work in designing the proper use of the city space [12]. Consequently, cities across the globe have lost considerable amounts of their adjoining quality farmland to urbanisation and have replaced green spaces with built-up areas, unproductive parks and decorative gardens. According to Pothukuchi and Kaufman [12], one of the main reasons for the food systems no longer being linked to the built environment are issues of hygiene, food safety and the proper use of city space. In fact, allocating urban plots for food production is a challenge because of many competing uses in cities and the high value of land [13]. Moreover, there remains controversy about the real economic impact of urban agriculture [14] and its function as a prime basis for food supply for urbanites [15]. Some even claim that urban agriculture contributes only insignificantly to total food production [16] and is economically not viable [17]. Irrespective of the differences in opinion about the place of urban agriculture, it is still on the rise globally [18,19] and is contributing towards alleviating poverty and survival [20,21] of more than 800 million people with 200 million of them being urban farmers [22].

The rising trend in urban agriculture practices is linked to its multifunctionality, which allows its scope to be defined in many different ways, giving rise to numerous definitions in the literature. They differ according to market specifications, production function and spatial boundaries [23]. Some offer a global perspective, while others are based on specific local contexts. Ultimately, urban agriculture should contribute towards making cities more sustainable. This review focusses on urban agriculture from a developed country perspective where no comprehensive policies exist to govern such activities. Consequently, the following definition is used: Urban agriculture covers agricultural activities located in (urban) or on the fringe (peri-urban) of a city or town, which reflect the local context and aim to enhance urban sustainability.

Currently, there are numerous activities related to urban agriculture. For example, community-supported agriculture, food co-ops, high-value crop production, direct sale to city dwellers and on-farm non-agricultural diversification measures are flourishing in many American and Canadian cities [24]. Local authorities in the UK and Germany are providing allotment sites to residents within the boundaries of the cities, so that they can engage in agricultural practices [25]. The benefits of urban agricultural practices are also numerous. They include the opportunity to locate production closer to consumption with shorter food supply chains as well as social benefits related to the provision of fresh food, increased mental health and improved local employment [24]. Economic benefits due to high yields in small spaces are also possible [26], particularly when technological innovations, such as hydroponics, aquaponics and vertical systems are used [27]. However, some of these projects may be associated with high running costs (e.g., for lighting, heating and running water) as well as require human surveillance and comply with health safety requirements, which are critical for their economic viability [28]. Urban food production adds to the multifunctionality of the urban fabric with a wide range of cultural (e.g., recreational, visual quality and cultural heritage) and ecological (e.g., nutrient recycling and use of urban waste, biodiversity conservation, creation of greenbelts and reducing the

impacts of climate change) benefits, all of which help urban communities and society as a whole [29]. Furthermore, city planning needs to adopt localised food production due to uncertain markets and trade, increasing prices for staple essentials, mounting concern about global warming, and availability of fossil fuel [30].

All of these issues contribute to the current concerns of urban sustainability. Moreover, the food system of a city contributes to its ecological footprint, including through food miles, energy use, organic waste recycling and production methods used, with urban agriculture reducing environmental impacts. With sustainability becoming an increasingly important concept in urban planning, localised food systems should be encouraged and facilitated [13]. Therefore, urban agriculture should not be excluded or viewed negatively, but rather endorsed as a way to improve city sustainability [31].

Such considerations are equally valid for Australia—a country ranked 24th among the most urbanised nations globally, with 89% of its population living in cities [32]. On the other hand, due to international migration, its population has been growing steadily. Rapid population growth presents a tremendous challenge to retaining and protecting vital agricultural land. Although food security has not been a prime political issue in Australia yet, it is currently becoming an area of concern, particularly for the socially disadvantaged within society [33]. For example, within a 12-month period, 15% of Australians have experienced food insecurity, and 3 in 5 of these people face this at least once in a month [34]. This prompts a consensus amongst academics and policymakers that urban agriculture is a practically feasible option for increasing city sustainability in terms of food and should be a part of the planning process for state and local governments [35]. However, in Australia, little scholastic attention has been given to consolidating urban agriculture into policy and planning [36], and globally, only a few studies have argued its contribution to food security [37].

The objective of this research review is to present a framework that will assist in integrating urban agriculture into city land-use policy and planning—an approach that could be replicated in a wider Australian context as well as elsewhere. An outline of the place of food production in the urban planning agenda is first presented, followed by an analysis that looks specifically at Australian practices. This leads to the proposed new framework for incorporating urban agriculture into city development and a discussion of its feasibility, implementation challenges and the role planners can play.

## 2. Food Perspectives in Urban Development Planning

Traditionally, planning addressed all basic essentials—air, water and shelter, except for food. In the urban planning context, it is generally viewed that food systems are mainly a rural issue and should not be part of its agenda. However, two prime arguments exist against such justification. Firstly, urban (city or local) food systems do have significant effects on other areas of valid interest to planners, such as land use, transportation, economic development, local employment, provision of energy and water, air pollution, public health and social justice. Secondly, seeing food production as a rural activity dismisses the potential importance of urban agriculture, particularly as farming in and around the cities has always existed [4,38]. In fact, throughout human history, city-based agriculture has been an inherent part of the livelihood of urban people [31,39], and only in more recent times has it declined in importance. However, now it can reconnect people to the urban (city or local) food system by facilitating local production close to communities [10]. The renewed attention to urban agriculture since the late 1980s and early 1990s manifests its significance for making rapid urbanisation all over the globe more sustainable. In the context of current development, introducing urban agriculture would be a viable means of achieving sustainability that addresses structural changes brought about by globalisation to communities, their food systems and quality of life for urbanites [40]. That is why some urban designers have been attempting to re-image 'the city as a farm' [41].

Cities now provide habitat for more than half of the world's population that needs to be fed [42]. In developed countries, the 20% wealthiest households spend between 6.5–9.2% of their income on food, whereas the poorest 20% of households spend 28.8–42.6% [43]. This figure is almost double in developing countries, where poor people in the city spend 60–85% of their budget on food [44].

This has drawn a renewed interest both in developing and developed countries for food to be a vital and integral part of the urban planning process. Issues of rapid urbanisation, food security, price surges, land conflict and climate change are justifications for planners to consider food planning seriously [40].

Urban food production needs to be prioritised as part of the vital city infrastructure, in the same way as roads or sewers are regarded an integral part of the system within built environments [45,46]. Considering the importance of urban food production, a Policy Guide on Community and Regional Food Planning was adopted by the American Planning Association in 2007 [47]. However, similar policy guidelines have not yet been drawn up in Australia. Planning of urban agriculture in Australia is at a very early stage, with no common strategy or policy provisions in place. On the other hand, there has been a revival in interest in community gardens [48], local food markets [49] and city farms in industrialised countries, including in Australia [50]. This indicates a grassroots swell towards reconnecting to the land, developing more green productive spaces and shortening of the food production systems. Many community groups and individual city dwellers are looking for local opportunities to grow food within the urban boundaries to which the planning system needs to be in a position to respond.

## 3. Urban Agriculture in Australia—Planning Tools and Legal Framework

The two main planning tools to manage land use and development in Australia are strategic land-use plans and land-use controls [51]. Strategic land-use plans cover broad land-use allocations for the future and are governed by the respective State governments. In addition, each local council controls land use within its area by identifying suitable land for development through zoning and by setting a standard to control this land use [52]. In Australia, urban planning policy has given very minimal consideration to supporting urban agriculture [36,40,53]. Historically, around the 1880s, cities in Australia started urban agriculture in the form of suburban food production, but due to changes in land-use policy, it soon began to disappear [54]. However, a number of regulatory measures have recently been adopted to restrain development on potential agricultural land on urban fringes [8]. At local levels, agriculturally sensitive urban designs have begun to be incorporated. For example, Melbourne's planning strategy emphasises the need to protect high-quality agricultural land together with undeveloped urban land for food production in and around the city [55]. The City of Brisbane in Queensland has taken the lead in facilitating access to fresh food in the urban environment, supporting the establishment of community gardens, providing places for farmers' markets and conserving agricultural land for food production [56]. Perth's planning document 'Directions 2031 and beyond' also emphasises the need to protect the agricultural land in urban fringes and recognises the importance of local food production [57].

Urban agriculture consists of multifunctional activities, which include production, transportation, processing and sale of food as well as utilisation and management of waste [58]. Because of this multifunctionality, urban agriculture can occur through various land uses, such as community gardening, roof top gardening, verge gardening, animal keeping, composting and farmers' markets. Land-use planning can reinforce, inspire, regulate or impede such urban agriculture practices. Cities can directly or indirectly influence these different urban agricultural practices through policies, zoning arrangements, programs and laws. Local councils can utilise planning laws to regulate both public and private land to foster urban agriculture. They have the power to use public land that can be licensed or leased and through zoning, they can dictate how any land can or cannot be used [59]. In most countries in the developed world, agriculture in the city is not recognised as an independent land-use category in municipal zoning plans [60]. There are, however, some exceptions, such as in Chicago [61], Vancouver [62] and European allotment gardens [63], where urban agriculture is regulated at various levels as a land-use category with some restrictions. Planners and municipalities have also developed some strategies such as the fringe plan for Copenhagen [64] and the agriculture buffer zones in the USA [65] to protect peri-urban agriculture. In fact, zoning is the main driving force behind city planning, and through it, the local government controls how its 'built-environment' is developed. In this case, planning law can be applied

for appropriate zoning and to control development in private land that mandates allocating space for local food production both in existing and new areas of the cities [66].

A study done by Pires [36] to analyse how local government within most Australian capital cities plans, encourages and regulates urban agricultural practices found numerous potential urban agriculture-related policies, strategic plans and regulations. This study revealed many encouraging signs of urban agricultural practices around Australia and noticed that councils had started to grapple with issues such as community gardening, green roofs, food production, animal keeping, composting, school gardening and farmers' markets. School gardening, farmers' markets and community gardening are now gaining popularity in Australia. Currently, there are 1630 primary schools, high schools and early learning centres that have been implementing kitchen garden programs nationwide [66] and more than 180 farmers' markets operating regularly in parks, plazas, school yards, showgrounds, community sheds and other venues across Australia [67,68]. In addition, more than 600 community gardens have been established, often supported by local councils [37,69,70]. However, councils are yet to have a substantial document dealing holistically with all aspects of urban agriculture [36]. Urban agriculture practices in Australia are frequently hampered by complex regulatory systems that hinder rather than facilitate their proliferation [64]. However, policy and planning mediations have an enormous potential impact in the promotion of urban agriculture interventions [71].

## 4. Integrating Agriculture into Urban Development Planning—A Framework

Urban development planning is currently adopting different progressive approaches, such as ecological models, new urbanism, collaborative and communicative models, city perspectives and new life models. Each of these provides specific ways and connections that could facilitate and stimulate the integration of urban agriculture. Planning for urban agriculture needs to go through a three-step process. The first step is to ensure legal provision through policy formulation which delivers the planning policies, regulations and legislation as the framework to regulate and guide urban land use for agricultural activities [72]. A second step is to establish official bodies to reinforce polices, programs, strategies and action plans. The third step is to identify, allocate and designate land according to availability and accessibility guidelines into master plans, structural plans and land-use zoning with provisions for ensuring tax incentives, tariffs and promotion of urban agriculture [73].

Tools such as site plans, master plans, local plans, neighbourhood plans and subject plans all serve to guide public safety, movement and transportation, community and individual health, and the use of private and public land [72]. However, they do not specifically address food security. The most important issue for urban food production is its official recognition as urban land use, security of tenure, as well as access to land and other resources [13]. Access to land is especially relevant for marginal and minority groups, and this could be mitigated by offering more publicly-owned open space for community gardens [74]. As not all city areas are well-suited for growing food, availability of land based on biophysical factors for urban agriculture could be identified by developing land-use inventories and land suitability analyses using geographic information system (GIS) technologies [42]. Access to sunlight is an important factor to be considered, particularly within the context of new construction and tree growth [75]. Water supply is also a consideration not only for crop production but also to clean and even process fruits and vegetables on site. Further considerations include resource availability, transportation systems, market connections and waste disposal systems [76].

A conceptual framework for integrating urban agriculture into planning and enhancing city sustainability is presented in Figure 1. It builds on the three pillars of the sustainability concept, namely, social (represented by food security and nutrition, poverty alleviation, improved health status, social cohesion and community building), economic (represented by income and employment generation, local development and enterprise) and environmental (represented by providing urban greenspace, reduction in the ecological footprint and enhancing urban habitat's biodiversity), allowing urban agriculture to flourish in the city. This requires land use and planning to come together to address

current sustainability challenges and respond to the factors that are already driving the presence of urban agriculture in the city.

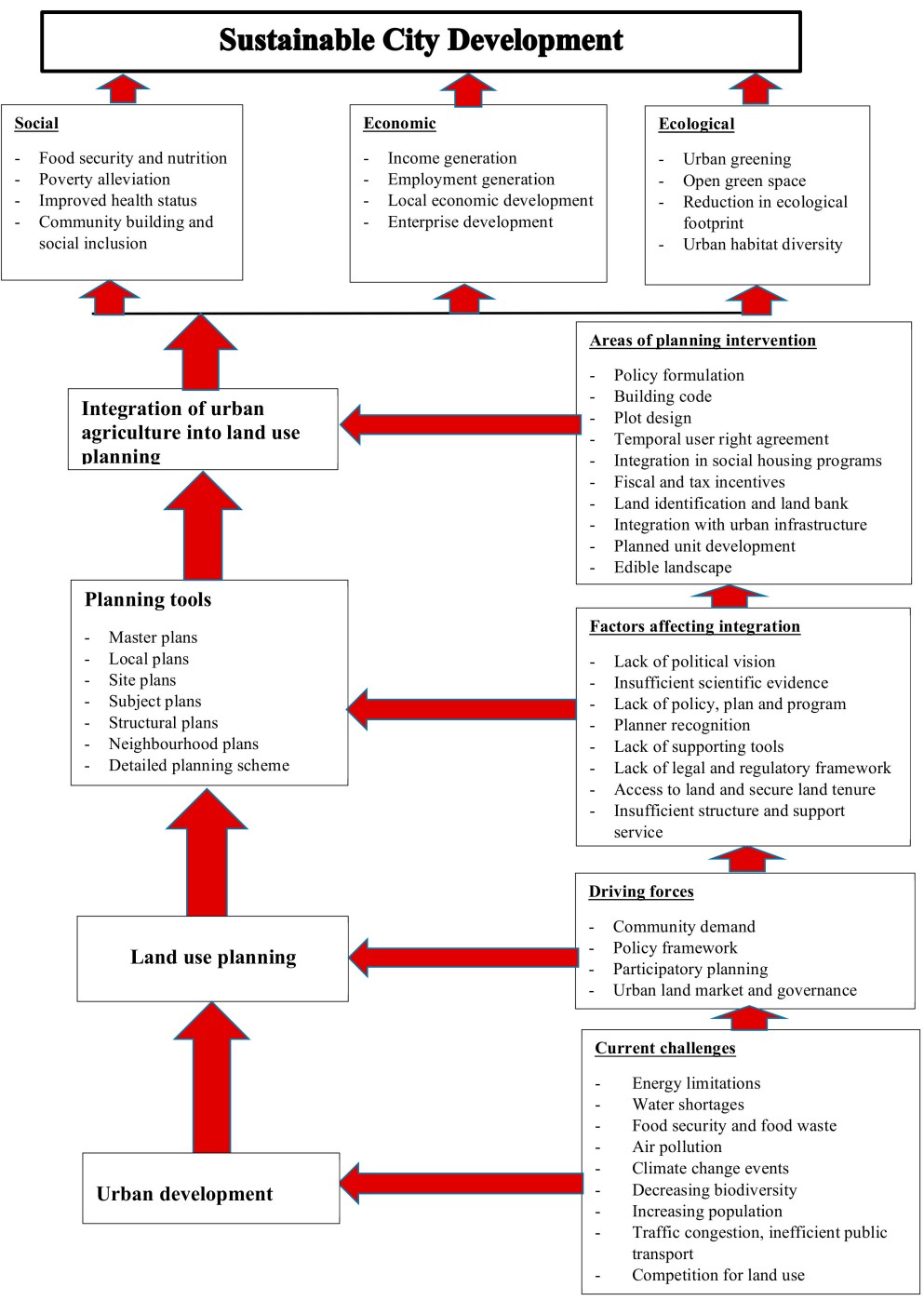

**Figure 1.** Conceptual framework for integrating urban agriculture into sustainable city development.

## 5. Framework Feasibility and Challenges to Implementation

Urban agriculture can play a vital role in meeting many of the food security challenges in the city. By doing this, it also helps to build greater resilience of the urbanites to climate change and other environmental and socioeconomic perturbations. However, there are significant policy barriers and external challenges including climate, biophysical, social, economic, cultural, regulatory and political,

which affect the development and wider adoption of these practices (Figure 1) [37,77]. The policy barriers and challenges are likely to continue to counteract the adoption of a proper framework for embedding urban agriculture in the planning of our cities unless a concerted effort is made towards sustainable city development.

The main constraints of urban agriculture planning can be categorised as being related to perception, regulatory and technical issues. According to Lovell [13], planners' perceptions are an important factor limiting urban agriculture integration. This is in line with earlier findings from a USA survey that attempted to understand why planners have not engaged in urban food systems, including urban agriculture [12]. Apart from the fact that planners did not perceive urban (city or local) food systems falling jurisdiction, they felt unqualified and did not have adequate knowledge to deal with the topic, particularly with its technicality and above all, they had not noticed any difficulties with the existing food system [12]. However,, they acknowledged lack of funding and linkages with community groups to support urban agriculture.

Further obstacles to the widespread adoption of urban agriculture activities can be inadequate infrastructure and other support services [42,52,78]. Intense competition for land use, due in large measure to a greater profitability of commercial enterprises, is another barrier to urban agriculture [44]. Growing food in urban and peri-urban areas has the added complexity of actual and perceived health risks [78]. Moreover, urban agriculture requires bringing together social, cultural and ecological knowledge and expertise.

For greater integration of urban food production into mainstream activities, many [22,60] suggest that the first step is to ensure conceptual clarity of the term 'urban agriculture' so that respective national and local governments can develop a regulatory and policy framework which addresses the potential risks related to such activities. The integration framework depicted in Figure 1 identifies the main driving forces of urban agriculture, the challenges faced in its integration into planning, available planning tools and areas of intervention for integrating urban agriculture in order to make cities more sustainable. However, the feasibility of this framework will depend on political will, regulatory support, planner perception and stakeholder engagement. "Feeding the city in a sustainable fashion—that is to say in a manner that is economically efficient, socially just and ecologically sound—is one of the quintessential challenges of the 21st century" [79] (p. 18), and this cannot be achieved without a visionary political commitment [80]. At present, urban agricultural practices are mostly driven by inadequate or unavailable support services to urban dwellers. With better resourced support services, greater economic benefits and larger-scale adoption, these barriers may start to be overcome [81].

In comparison to Australia, many places around the world have already been able to build urban agriculture into the city fabric. For example, Quito, the capital of Ecuador has had a successful urban agriculture program since 2002, which has helped to establish more than 1700 gardens with more than 400 tonnes of annual crop production [82]. Toronto and Montreal in Canada also preserve agricultural rich areas within the metropolitan regions to shorten food supply chains in providing safe and nutritious produce [28]. In these cases, spatial planning plays an essential role in protecting agricultural land from expansion in the built environment. Many African cities have advanced agricultural sectors, an example being Dar es Salaam, Tanzania, where urban agriculture is mainstreamed in land-use strategies [22]. Allocation of allotments for urban agriculture is compulsory in the planning process for German cities [25]. Urban agriculture encourages grassroots participation and brings community together, particularly in difficult times, as was the case for Portuguese cities, including its capital Lisbon, after the 2008 Global Financial Crisis [81]. In the USA, urban agriculture is seen as a way to contribute and maintain the green environments in the city, and its multifunctionality supports biodiversity and other ecological functions while improving the social fabric and health of communities [29]. Its economic benefits can also be substantial as in the case of Milwaukee in Wisconsin [29]. The entire city of Portland in Oregon is an edible landscape which has helped its transitioning to sustainability [29]. The American Planning Association also recognises the importance of incorporating urban agriculture into land-use planning [29]. Havana, Cuba offers an impressive example of urban agriculture not only in allocating 30,000 hectares of land within and on the fringe of the city but also in the use of

organoponics—raised beds with quality soils that can be put in place on any surface [29]. Asian cities are considered the birthplace of urban agriculture [22], with many of them, including Shanghai in China, still preserving productive lands for growing fresh food [29].

As urban agriculture is not without risk, adequate planning and regulation efforts need to be put in place to prevent soil, water and air pollution from pesticide and fertiliser use, while allowing for a large choice of production methods—"home gardens, community gardens, allotments, school gardens, balcony and rooftop gardens, working at various scales, from balconies to large farms at the urban/rural interface" [81] (p. 94). Another major concern is competition for land and conflicts which can occur with other commercially more profitable activities [44,81]. Other arguments against urban agriculture relate to health risks, particularly the presence of heavy metals and persistent organic pollutants, such as polychlorinated biphenyl used as pesticide extenders, flame retardants, coolants and plasticisers. Toxic accumulation can also occur because of proximity to other areas of non-agricultural use, such as industrial zones, road arteries with heavy traffic or landfill waste disposal facilities [81]. All these concerns reinforce the importance of proper planning processes to mitigate risk given the fact that cities continue to represent "some of the more polluted environments on earth" [81] (p. 96).

The proposed framework for integrating urban agriculture in the process of transitioning towards more sustainable cities brings together all aspects of what are very complex processes. Although none of its elements are necessarily new, they are presented in a novel logical way that allows a complete picture to be built, presenting the challenges at each stage of the planning process and each providing the tools and interventions that can resolve or avoid potential tensions.

It is a matter of fact that in many ways, planners are key actors to for successfully facilitating and promoting urban agriculture activities. Planning constraints can be eliminated by reframing the role of the planner.

## 6. Integrating Urban Agriculture—The Role of the Planner

As a profession, the aim of planning is to ameliorate the health and welfare of communities and individual people through the rational arrangement of resources, facilities and land [83,84]. Integrating agriculture into urban planning has become a burning topic both in developed and developing countries around the world [15]. Australia should also be part of this conceptual change. Urban planners could play a major role in framing planning to incorporate urban agricultural features for addressing some of the current issues and future challenges, such as preservation of agricultural land within the city boundaries, enhancing access to quality fresh food by changing zoning and land use, incorporating local food activities into economic development and, overall, addressing the food system's environmental impacts [12,71,74]. There is a significant opportunity for planners to enhance city sustainability by including urban agriculture into the planning agenda at all levels—from the master plan to the plans for individual sites and neighbourhoods. Planners can contribute towards better cities and healthier communities by making urban agriculture an aspect of their practice through its integration in urban infrastructures, planned unit developments, housing projects and by preserving or establishing edible landscapes [85]. The conceptual framework shown in Figure 1 points out additional key issues in which planners should play a role to facilitate urban agriculture, such as policy formulation, building code, plot design, formulation of temporal user rights agreements, integration in social housing programs, fiscal and tax incentives, land identification, and establishing land banks where planning can intervene to promote urban agricultural practices.

A particularly valuable feature of the role of the planner is in providing better opportunities for the socially vulnerable within society, where urban agriculture could foster food justice. Horst [85] points out how urban planners can support food justice through a range of initiatives which include prioritising urban agriculture in any long-term planning, paying extra attention to facilitating such activities through mutually respectful relationships with participating people from diverse backgrounds, targeting investments to benefit socially disadvantaged communities, increasing the permanently available land and removing threats to displace agricultural pursuits with other land uses.

Recently, there have been indications in some countries, notably the USA and Canada, for planners to become involved to a greater extent in promoting urban agriculture by offering discounts in fees and taxes, removing legal barriers, ensuring long-term or permanent access to land, providing staff and resources and prioritising urban agriculture in the planning agenda [86,87]. With direct public policy support and institutional recognition, urban agriculture can successfully be incorporated into the planning and development goals of making cities more sustainable where planners are in a position to make the strongest formal contribution to policy reform [88,89]. As Dubbeling et al. [89] (p. 12) note, a sustainable and resilient urban food system requires "the creative use of available policy and planning instruments (infrastructure and logistics; public procurement; licences; land-use planning, etc.); and the involvement of different government departments and jurisdictions (local and provincial) and new organisational structures at different scales (municipal, territorial, etc.)".

Although the land requirements and productivity of urban agriculture would vary from place to place and would also depend on the methods used (e.g., use of fertilisers, organic or biodynamic growing of plants), it is possible to provide some estimates of the food calorific potential of a garden bed. As Australian cities have a comparable large number of sunny days as in Fort Collins, Colorado, USA and the diets in both countries are very similar, it may be possible to draw comparisons with this American case study which refers to a raised bed garden [90]. Table 1 shows the annual nutritional contribution of the plants grown in similar conditions on a 10 m$^2$ raised bed garden based on a daily intake of 2000 calories for an average adult. These data, however, need to be put into perspective as to what kind of a diet and lifestyle the urban population is adopting. For example, vitamin D can easily be supplied through exposure to the sun. Similarly, the protein intake can be limited only to the essential amino acids as the others can be produced by the human body itself. Hence, the respective percentages of the recommended intake are only ballpark estimates.

**Table 1.** Annual nutritional production from a 10 m$^2$ raised bed garden.

| Product/Nutrient | Total Production | Percentage of Recommended Annual Intake |
| --- | --- | --- |
| Vegetables (kg) | 19.4 | 17.2% |
| Micronutrients | | |
| Folate (µg) | 5560 | 3.8% |
| Phosphorous (mg) | 6499 | 1.8% |
| Protein (g) | 236 | 1.3% |
| Riboflavin (mg) | 12 | 1.9% |
| Saturated fatty acids (g) | 8.6 | 0.1% |
| Vitamin A (IU) | 278,551 | 15.3% |
| Vitamin B6 (mg) | 29 | 3.9% |
| Vitamin C (mg) | 4915 | 22.4% |
| Vitamin D (IU) | 0 | 0% |
| Vitamin K (µg) | 7272 | 24.9% |

Source: Calculated from [90].

Research in the 1970s [91] estimated that one person can be sustained on a plant-based diet on a 720 m$^2$ block, while in the 1980s, Robbins [92] calculated a vegan diet to require 1/6th acre or 676 m$^2$, while a vegetarian diet, which incorporates eggs and dairy, needs 1/2 acre or 2023 m$^2$. By comparison, a standard American or Australian diet requires 3 acres or 12,141 m$^2$ per person [92]. It is obvious that urban agriculture can assist food security, particularly for plant-based options. One square kilometre of urban agricultural land can indeed feed 1482 people on a vegan diet.

In addition to the food calorific potential, urban agriculture has numerous other benefits—economic, social and environmental. Table 2 lists some of them. There are, however, potential hazards, such as elevated levels of phosphorous and nitrogen in the soils, if compost and fertiliser management is not done properly [90].

**Table 2.** Examples of nonfood-related benefits from urban agriculture.

| | |
|---|---|
| Economic | Reduces household food expenses through food production; creates jobs; increases property values |
| Social | Promotes food education and awareness; fosters civic engagement; provides recreational activities; generates aesthetic values and beauty |
| Nature preservation | Reduces land conversion for agriculture and allows reuse of currently irrigated lawns in some regions; increases nutrient recycling opportunities, including recycling of household waste; better biodiversity and provides habitat for some species; reduces transportation and storage of carbon; offsets agricultural water use if captured rainwater can meet garden needs; conserves genetic diversity |

Source: Compiled from [90].

The role of the urban planner is to take all these complex aspects into consideration when integrating urban agriculture into city planning and provide access to appropriate land. Developing urban agricultural enterprises depends highly on availability of land, which influences the ability to make capital investments, crop selection, labour force and market access. In addition to land tenure, local land-use practices, policies and prices can influence the viability of an operation. Site selection for urban agriculture should be based, among other factors, on:

- Accessibility and availability of suitable land with length of tenure;
- Soil quality, drainage and land-use history;
- Noncontiguous lots;
- Access to water;
- Location in a growing area—highly visible and with proximity to public transport;
- Strong neighbourhood support;
- Low levels of traffic congestion;
- Adequate hours of full sunlight.

Most importantly, considerations should be given to the surrounding built and natural environment, and this is where the role of planning is essential. Planning will need to make some tough decisions and compromises between the overlapping needs and wants within the contested urban land space. This is a new challenge that will need to be adequately addressed given the overall benefits of finding space for urban agriculture in the 21st century's cities.

## 7. Conclusions

There are growing indications that awareness about the importance of the future of urban food systems is rapidly increasing as the world faces the challenges of providing reliable access to nutritional and fresh food options. Food security is being challenged by growing urban populations, unfavourable climate change and extreme weather events as well as market and trade variability. In 2015, the Milan Urban Food Policy Pact [93] was launched, and it has attracted 184 signatories to date. The Pact confirms that urban agriculture can significantly augment local food supply, which can in turn help to maintain a sustainable urban life and urban resilience. From Australia, only Melbourne has joined the Pact, which calls for every city to tackle the problems related to building of a reliable, safe and sustainable urban food system. On the other hand, the experience in cities such as Portland, Havana, Quito, Berlin, Copenhagen and Toronto, shows that planning can substantially improve the possibilities for urban agriculture within the city limits, including its peri-urban areas.

One of the main points by Dubbeling et al. [89] that brings an additional component to the need for integrative planning is the fact that urban development and food systems are coupled to activities outside the city boundaries because of the way in which metropolitan areas draw resources and affect their surroundings. By providing options for urban agriculture within the territory or at the fringe of the cities, more sustainable relationships can emerge between urbanites and the provision of food for

their consumption. This can help to achieve broader sustainable development goals, including the current targets of zero hunger and curbing malnutrition. Planning for urban agriculture, however, should be done in a way that maximises its benefits and minimises any potential risks.

The conceptual framework (Figure 1) presented in this review could serve as a guideline to successfully integrate urban agriculture into land-use planning to make cities more sustainable in places where this still needs substantial effort. This integration framework requires policy support, institutional recognition and strong political will to bring sustainable urban agriculture into city planning. Although planning has commenced the journey of adopting the sustainability agenda for Australian cities, there remains much more to be done to achieve a collaborative and integrative approach to urban agriculture by planners and make this an essential aspect of the vision and design for any level plans.

**Author Contributions:** A.H.S. conceived and wrote the paper; J.F.B. and D.M. contributed to the manuscript.

**Funding:** This research received no external funding.

**Conflicts of Interest:** The authors declare no conflict of interest.

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
