# Peer review of "A Framework for Integrating Agriculture in Urban Sustainability in Australia"

_urbansci, doi:10.3390/urbansci3020050_

Reviewer 1 Report

Review report

Article: A framework for integrating agriculture in urban sustainability in Australia

(Manuscript ID: Urban Science--466522)

This paper discusses how urban agriculture can be integrated into city land-use policy and planning. The case study of the analysis is the Australian context. The main contribution of the paper is the suggestion of a framework for incorporating urban agriculture into city development.

The topic under discussion is interesting and the paper clearly defines it.

Nevertheless, in my opinion, there are two weaknesses the authors must deal with in order to increase the interest of paper in particular for non-Australian nationality readers.

Main comments.

1. The analysis of the Australian experience should include the comparison with that of other nations. In so doing, its importance could better appreciate by non-Australian nationality readers.

Since the current version of the Conclusions section is very lacking in suggestions and penalises the balance of paper structure, I would like to see in this section the discussion of the findings of the paper in relation to the experiences gained in Europe, USA or other countries.

 2. The second weakness of the paper is the lack of "(...) conceptual clarity about the term" urban agriculture "(lines 256-257).

Despite the statement, the definition was not reported and at line 266 is wrote "(...) Consequently, the following definition is proposed, which captures the essence of the urban agriculture integration framework in Fig.1:" (???)

What is the proposed definition?

Please, provide it and discuss its appropriateness extensively.

In summary, my overall evaluation of the paper is positive; however, I think the paper needs to be revised.

Author Response

REVIEWER 1

Thanks for the useful comments which helped improve the quality of the manuscript. Below is our response to these helpful recommendations:

Comment 1. “The analysis of the Australian experience should include a comparison with that of other nations. In so doing, its importance could better appreciate by non-Australian nationality readers.

Since the current version of the Conclusions section is very lacking in suggestions and penalises the balance of paper structure, I would like to see in this section the discussion of the findings of the paper in relation to the experiences gained in Europe, USA or other countries."

Response to Comment 1: A new section has been included on pp. 7-8 which provides an international perspective on the issues of urban agriculture; this international perspective is also brought in the Conclusion on p. 9.

Comment 2. “The second weakness of the paper is the lack of "(...) conceptual clarity about the term" urban agriculture "(lines 256-257).

Despite the statement, the definition was not reported and at line 266 is written "(...) Consequently, the following definition is proposed, which captures the essence of the urban agriculture integration framework in Fig.1:" (???)

What is the proposed definition? Please, provide it and discuss its appropriateness extensively."

Response to Comment 2: Our apologies for the technical error in formatting the text which made the definition look as a subheading in the originally submitted manuscript. The revised version of the paper includes the definition of urban agriculture together with a discussion of its possible perspective earlier in the paper on p. 2.

General response: The English of the text also been checked and the level of similarity has been significantly reduced.

Reviewer 2 Report

Dear Authors,

thank you for sharing your work with the journal, which asked me for a review of your submitted article. I am happy to see your work, but see some key aspects I would like to highlight. All other comments are indicated in the uploaded document via comments.

- For a review article the presented depth of literature review is not enough. You need more and more detailed references for your review and furthermore it would be positive to review the literature with a critical lense. The review and the overall article shows a biased and very positive picture of urban agriculture neglecting all constraints, conflicts, disadvantages, etc. which might occur.

- The definition of urban agriculture should be positioned at the beginning. For the majority of the text it is not clear, what you mean with urban agriculture. Sometimes you name urban gardening elements, sometimes hydroponics/aquaponics, sometimes farmer markets, but only in the end of the document the readers get aware of the definition. This makes the review part of your paper difficult to read.

- I have some concerns about the framework of integrating urban agriculture into urban development planning. It is indicated as a review article, but the review is on a medium to low level, while you put some work into the framework. From my point of view it sounds more like a handbook/manual than a scientific paper. Furthermore, I do not really see any new issues in the framework.

Best regards

The reviewer

Author Response

Thank you for your insights and suggestions to improve the quality of the manuscript. We believe we addressed all of your concerns as follows:

Comment 1. "- For a review article the presented depth of literature review is not enough. You need more and more detailed references for your review and furthermore, it would be positive to review the literature with a critical lens. The review and the overall article shows a biased and very positive picture of urban agriculture neglecting all constraints, conflicts, disadvantages, etc. which might occur."

Response to Comment 1: A new section is now included which discusses the negative aspects of urban agriculture (see p. 8).

Comment 2. "- The definition of urban agriculture should be positioned at the beginning. For the majority of the text, it is not clear, what you mean with urban agriculture. Sometimes you name urban gardening elements, sometimes hydroponics/aquaponics, sometimes farmer markets, but only at the end of the document the readers get aware of the definition. This makes the review part of your paper difficult to read."

Response to Comment 2: The definition of urban agriculture is now brought forward to p. 2.

Comment 3. "- I have some concerns about the framework of integrating urban agriculture into urban development planning. It is indicated as a review article, but the review is on a medium to a low level, while you put some work into the framework. From my point of view, it sounds more like a handbook/manual than a scientific paper. Furthermore, I do not really see any new issues in the framework."

Response to Comment 3. A new paragraph has been added on p. 8 which explains the contribution the newly proposed framework makes in bringing all aspects together of a process which is highly complex and new for Australia.

General response: The English of the text also been checked and the level of similarity has been significantly reduced.

Round  2

Reviewer 1 Report

Article: A framework for integrating agriculture in urban sustainability in Australia

(Manuscript ID: Urban Science--466522 - 2nd revised version)

The revised version of the paper is now in good shape for publication. Just a few minor things reported in the attached file.

Author Response

We have accepted all of your recommendations and updated the manuscript as you advised.

Reviewer 2 Report

Dear authors,

many thansk for taking the reviewers' comments into account for improving your work. I am very much in favour of your current work and think that it is a good paper and adequate for publication in the journal.

Just three minor remarks:

1) References should be included when naming the various forms of UA (ll. 84ff.)

2) Economy of technological innovations:I am not convinced by the statement that (very) high investments and rather high running costs (e. g. for aquaponics the electricity costs for heating, lightning, and pumping water) - partly together with high demands for human surveillance of the fish and technical equipment - increases economic performances. We have several examples of these high-tech inner-city projects, which have failed (urban farmers in Basel and The Hague for example).
Thus, I would like to see it a bit more critical here and not link the statement only to one reference.

3) Figure 1: Improvements for Figure 1 needed: Layout, readability (some last lines of boxes are cutted and not good to read)

Best regards

Author Response

(The authors gave the same response as above.)
